# Morphologic Alterations of the Sternoclavicular Joint following Ipsilateral Clavicle Fractures

**DOI:** 10.3390/ijerph192215011

**Published:** 2022-11-15

**Authors:** Malik Jessen, Marc Schnetzke, Stefan Studier-Fischer, Paul Alfred Grützner, Thorsten Gühring, Felix Porschke

**Affiliations:** 1Department of Trauma Surgery, Clinic Rechts der Isar, Technical University of Munich, Ismaninger Strasse 22, 81675 Munich, Germany; 2German Joint Center Heidelberg, ATOS Clinic Heidelberg, Bismarckstr. 9-15, 69115 Heidelberg, Germany; 3BG Trauma Center Ludwigshafen, Heidelberg University, Ludwig-Guttmann-Straße 13, 67071 Ludwigshafen, Germany; 4Orthopedic Clinic Paulinenhilfe, Diakonieklinikum Stuttgart, Rosenbergstr. 38, 70176 Stuttgart, Germany

**Keywords:** concomitant injury, joint space width, dislocation, instability, subclinical, secondary damage

## Abstract

Rationale: To our knowledge, no study has investigated concomitant injuries of the sternoclavicular joint (SCJ) in acute clavicle fractures. The purpose of this study was to determine the effect of an ipsilateral clavicle fracture on the SCJ in a systematic computer tomography (CT) morphologic evaluation. Methods: CT scans in the axial and coronal plane of 45 consecutive patients with clavicle fractures were retrospectively analyzed. The scans were assessed regarding anatomic congruence of bilateral SCJs—joint space width (JSW); the position of bilateral medial clavicles (PC); and the non-fusion of epiphyses, arthritis, calcifications, and intra-articular gas. Results: The mean SCJ JSW was significantly different in the coronal (cJSW; 8.70 mm ± 2.61 mm in affected vs. 7.63 mm ± 2.58 mm in non-affected side; *p* = 0.001) and axial plane (aJSW; 9.40 mm ± 2.76 mm in affected vs. 9.02 ± 2.99 in non-affected SCJs; *p* = 0.044). The position of the medial clavicle showed a significant difference in the coronal plane (cPC; 14.31 mm ± 3.66 mm in the affected vs. 13.49 ± 3.34 in the non-affected side; *p* = 0.011), indicating a superior shift. Conclusion: Acute clavicle fractures may be associated with an enlargement of the ipsilateral SCJ space width and a superior shift of the proximal clavicle. Both morphologic alterations could indicate concomitant injuries of the SCJ as well as a potential increase in the risk of SCJ instability.

## 1. Introduction

The sternoclavicular joint (SCJ) is a synovial joint and the only bony connection between the upper extremity and the trunk. It allows for the guided lifting, lowering, and forward and backward movement of the shoulder, as well as its circumduction. Injuries of the SCJ are rare [1] and can be easily missed in the primary survey of a patient, especially in those who are polytraumatized [2,3]. SCJ injuries are clinically categorized into three stages: sprain, subluxation, and dislocation [4]. However, there is no radiological classification.

Only 3% of dislocations around the shoulder are represented by sternoclavicular joint dislocations [5], which can be either traumatic or non-traumatic and are divided into anterior and posterior.

Anterior dislocations or subluxations of the SCJ are more common than posterior dislocations and can result in functional impairment [6]. In the case of anterior subluxations and dislocations, the treatment may include reduction and rehabilitation, though some instances may require stabilization through surgical intervention and management [3,6].

SCJ injury leading to posterior subluxation or dislocation has life-threatening implications due to the mediastinal anatomy that lies posterior to the SCJ such as the trachea, esophagus, subclavian vessels, and parts of the brachial plexus [7,8]. Therefore, surgical intervention is usually required to not only reduce the dislocation but to also maintain stability with the use of plates and screws or tendinous grafts depending upon what anatomical structures have been damaged. In general, there is a current trend towards surgical rather than conservative management for symptomatic SCJ injuries [9,10]. SC dislocations associated with clavicle fractures are described in case reports [11,12].

In contrast, clavicle fractures are common and represent 8.6% of all upper extremity fractures [13]. The most widely used classifications for clavicle fractures are the Robinson, Allman, and Neer classification [14,15,16,17]. All classifications divide the fractures into lateral, midshaft, and medial. The Robinson classification subdivides all three fracture types. Thus, especially the midshaft and medial fractures can be more differentiated and described by using the Robinson instead of the Allman or Neer classification. Because clavicle fractures are often associated with high-energy trauma [18], it can be assumed that forces leading to a fracture of the clavicle might compromise the SCJ. The evidence for an impact of clavicle fracture on SCJ is weak and only case reports have been published [19,20,21,22,23,24]. In contrast to the complete dislocation of the SCJ, which can lead to severe complications and is diagnosed initially in most cases, subclinical SCJ injury might be missed due to the clinical focus on the clavicle fracture and concomitant soft tissue injuries [3]. To our knowledge, no study has investigated concomitant injuries of the SCJ in acute clavicle fractures. 

The purpose of this study was to determine the effect of ipsilateral clavicle fracture on the SCJ in a systematic computer tomography (CT) morphologic evaluation. We hypothesized that clavicle fractures cause CT morphologic alterations in ipsilateral SCJ compared with the non-affected side.

## 2. Materials and Methods

### 2.1. Study Population

We conducted a retrospective review of the institutional database of all patients who had a clavicle fracture and were admitted to our Level I trauma center between 2015 and 2019. We included all consecutive patients with clavicle fractures who underwent CT evaluation, including imaging of the shoulder and SCJ. The exclusion criteria were set as follows: (1) in CT detected old fractures or injuries of any side of the clavicles, sternum, SC, or acromioclavicular (AC) joints; (2) bilateral clavicle fracture; (3) scapula fracture; (4) clinically significant SCJ dislocation; (5) injury of ipsi- or contralateral AC joint; (6) degraded image quality due to motion-related artifacts; (7) missing any side of the clavicles, AC or SC joints in the scan; and (8) arm positions while scanning other than 0° abduction and 0° forward flexion.

### 2.2. Radiological Evaluation

For all patients, the same CT scanner (Aquilion 32; Canon Medical Systems, Tokyo, Japan) was used. CT scans were evaluated using the software AGFA IMPAX 6.5.5.1608. Each scan was analyzed in the coronal (c) and axial (a) imaging plane (2 mm slice thickness). The radiological evaluation was performed by a shoulder-fellowship-trained physician (author F.P.). The clavicle fractures were classified following the classification system developed by Robinson et al. [15]. The morphology of the SCJ was evaluated as follows: First, we measured the joint space width (JSW) of both SCJs. We defined JSW by the least distance between the clavicular and sternal corticalis (Figure 1, red arrows). Second, we evaluated the positioning of the medial clavicle (PC) to assess a possible translation in the SCJ after clavicle fracture (Figure 1, green arrows). To evaluate the PC, we created a tangent through the most cranial corticalis of the sternum (Figure 1, white line). Then, we measured the distance, orthogonal to the tangent, to the most cranial corticalis of the medial clavicle and defined this as PC (Figure 1, green arrows). The JSW and the PC of the affected and non-affected SCJ in each individuum were compared in both the axial (aJSW, aPC) and coronal (cJSW, cPC) imaging plane. Subsequently, to assess differences in the JSW and PC with respect to fracture location (lateral, midshaft, and medial clavicle fractures), the mean values from each fracture location were compared.

To investigate further changes in the SCJ after clavicle fracture, the SCJ was examined for the presence of intra-articular gas as a possible sign of trauma sequelae [25,26]. Finally, to exclude discrepancies in both SCJs due to degenerative changes, signs of osteoarthritis and calcifications were examined [27].

### 2.3. Statistical Methods

Statistical results were calculated with SPSS^®^ Statistics Version 24 (IBM, Armonk, NY, USA). For all values, descriptive statistics were applied using the mean, standard deviation (SD), and minimum and maximum values. The Kolmogorov–Smirnov test was used to test for normal distribution. A comparison of the variables was made using a t-test for normally distributed data and the Mann–Whitney U test for not normally distributed data. The McNemar test was used for matched-pair data. The Kruskal–Wallis test was used to compare not normally distributed variables when there were more than two groups. Statistical analysis was made with two-tailed p-values, and the alpha level was set at 0.05.

## 3. Results

### 3.1. Demographic Data

A cohort of 97 consecutive patients was identified in our institutional database. After applying exclusion criteria (Figure 2), 45 patients were included. Of these, 35 were male patients (78%) and 10 were female patients (22%). The mean patient age at the time of the initial trauma was 49.71 ± 18.45 years (range: 13–79 years). Reported causes of the trauma were: fall from a height (*n* = 11), motorcycle accident (*n* = 10), car accident (*n* = 9), bicycle accident (*n* = 8), fall in a domestic setting (*n* = 4) and occupational accident (*n* = 3).

### 3.2. Clavicle Fractures

A review of the CT scans demonstrated 4 (8.9%) medial, 31 (68.9%) midshaft, and 10 (22.2%) lateral clavicular fractures (Table 1) according to the Robinson classification [15]. In total, four (8.9%) medial and two (4.4%) lateral clavicle fractures were intra-articular.

### 3.3. SCJ Morphology

The mean JSW, measured as described above (Figure 1, red arrows), was significantly different in both the coronal (cJSW; *p* < 0.001) and axial plane (aJSW; *p* = 0.044) (Table 2). The mean difference for cJSW was 1.07 mm (14.02%) and for aJSW was 0.67 mm (7.93%). Therefore, a significant enlargement of the JSW in the affected vs. non-affected side of the SCJ was presented. The position of the clavicle showed a significant difference (*p* = 0.011) in the coronal plane indicating a superior shift of the clavicle after clavicle fracture (cPC; mean difference 0.83 mm (6.08%)).

After analyzing the axial plane, no significant difference (*p* = 0.274) was found (aPC; mean difference 0.38 mm (4.02%)) (Table 2).

Then, we compared the mean values (differences between the affected and non-affected side in mm) depending on the fracture location (Table 3). Although there was a slightly significant result for cJSW (*p* = 0.048), we were unable to find any overall differences in the mean values for the different fracture locations.

Intra-articular gas was found in two affected joints (4.4%), and was not found in the non-affected joints. However, these findings were not significant. The number of osteoarthritis cases did not differ between the affected and the non-affected side (11 (24.4%) in both groups). There were no significant differences found for calcifications (three (6.7%) in the affected vs. five (11.1%) in the non-affected SCJs).

## 4. Discussion

The results of this study confirmed our hypothesis that SCJ congruency is altered following an ipsilateral clavicle fracture. The SCJ space was significantly widened at the side of the clavicle fracture compared with the uninjured side. Our results showed significant differences for the PC in the coronal plane but not in the axial plane, so we concluded that the proximal clavicle tends to shift superior following clavicle fracture. Both a widened SCJ and a superior shift of the proximal clavicle may represent subclinical SCJ instability, which could lead to chronic instability and osteoarthritis [28]. Therefore, our findings describe morphologic changes in the SCJ in clavicle fractures, which may lead to clinical symptoms of the SCJ sometime after the initial trauma.

A CT morphological description of the SCJ width was previously performed in healthy patients [29,30,31].

Tuscano et al. [29] described, in 104 healthy patients, a SCJ space width between 0.0 and 5.7 mm by measuring three distances and taking the mean value for the calculation of the SCJ space width. They demonstrated substantial asymmetries between the right and left SCJ in some cases. In limitation, CT scans were only evaluated in the axial plane. De Maeseneer et al. [30] conducted a study including 66 healthy patients. They could not find any significant differences between left and right SCJ space widths either in the coronal or axial plane using the same measurement technique described by Tuscano et al. [29]. The same method was used by Cakmak et al., and they also found no significant difference between both SCJ sides in 73 healthy patients [31].

To our knowledge, no study has performed a structural analysis of the SCJ in a traumatic fracture of the clavicle. We found that the SCJ space difference was significantly altered with a larger SCJ space on the affected side with a mean difference of 14.02% in coronal and 7.93% in axial imaging plane (Table 2). In contrast to previous studies [29,30,31], our study did not examine a healthy patient population, and consequently, we attributed the SCJ space difference as an effect of the ipsilateral clavicle fracture.

The results showed a significant change in the PC in the coronal, but not in the axial plane of the CT scan (Table 2). We considered that the position of the clavicle tends towards a superior shift after ipsilateral clavicle fracture. In addition to the widened SCJ space width, the altered position of the ipsilateral clavicle may be regarded as a subclinical aspect of an SCJ injury. Typically, a routine examination of the SCJ tends to focus on anteroposterior translation to detect instability. Therefore, a superior translation of the clavicle is probably not identified. Therefore, an additional focus should be on superior instability.

Among the fractures divided into medial, midshaft, and lateral, we found no significant differences in JSW and PC (Table 3). JSW was slightly significantly increased (*p* = 0.048) in the medial compared with midshaft and lateral clavicle fractures. However, due to the small sample of medial clavicle fractures (*n* = 4), we did not attribute an effect to this finding.

Our findings of intra-articular gas were 4.4% (*n* = 2) in affected vs. 0% in non-affected SCJs. Intra-articular gas may indicate an injury. Due to the much higher prevalence of 8% [32], 33% [29], and 42% [31] of intra-articular gas in SCJ described in the literature, we assume that the findings of intra-articular gas in this study have no pathological significance. Our results concerning calcifications (6.7% in affected vs. 11.1% in non-affected SCJ) may also indicate avulsions or pre-existing degeneration at the SCJ. As calcifications were more frequent in non-affected SCJs and the findings are consistent with the prevalence of the age group for calcium pyrophosphate dihydrate crystal deposition (CPPD) [27], it can be assumed that these are non-pathological calcifications. Twenty-four percent of our patients showed signs of osteoarthritis, which is low concerning the mean age of 49.71 ± 18.45 years, as the literature reports 50–90% osteoarthritis for individuals over 60 years [33,34,35].

The strengths of this study are the practicability in clinical practice, as regular standard reconstructions for the cervical spine have been used for radiological analysis. CT is the imaging technique of choice and is most widely used for the evaluation of the SCJ and for observing additional injuries to the upper thoracic ring, especially with high-impact trauma mechanisms [36,37,38].

Nevertheless, there are some limitations to this study. It should be noted that a specific reconstruction for anatomical SCJ analysis would be more accurate, which would not be appropriate in clinical routine in the context of a polytrauma scan. In previous studies on SCJ, a CT neck was also used [29,30].

Only one experienced observer did the radiological evaluation. However, intra-observer reliability of the measurements was tested in a preliminary study in 40 randomly selected, non-traumatic, and healthy upper extremity/musculoskeletal patients receiving a chest CT resulting in intra-class correlation coefficients (ICC) values between 0.88 and 0.94. Our approach does not consider the super inferior and anteroposterior differences in joint space width, as described by De Maeseneer et al. [30]. The data of this study were collected retrospectively.

Furthermore, the statistical reliability of the JSW and PC results of the medial clavicle fracture cohort is weak due to a reduced sample size (*n* = 4). This limitation should be carefully considered in the interpretation of the results regarding the medial fractures.

Moreover, we did not correlate our radiological findings with clinical information such as tenderness or swelling. Consequently, there is no information about the SCJ in progress and whether subclinical aspects of an SCJ injury led to secondary degenerative changes. Further studies on the impact of a clavicle fracture should include a correlation between radiological findings and clinical outcomes.

## 5. Conclusions

Clavicle fractures were associated with an enlargement of the ipsilateral SCJ space and a superior shift of the medial clavicle, especially in medial clavicle fractures. These findings emphasize a focus on the SCJ during the initial diagnosis and treatment of the clavicle fracture to determine the subclinical aspects of a damaged ipsilateral SCJ.

## Figures and Tables

**Figure 1 ijerph-19-15011-f001:**
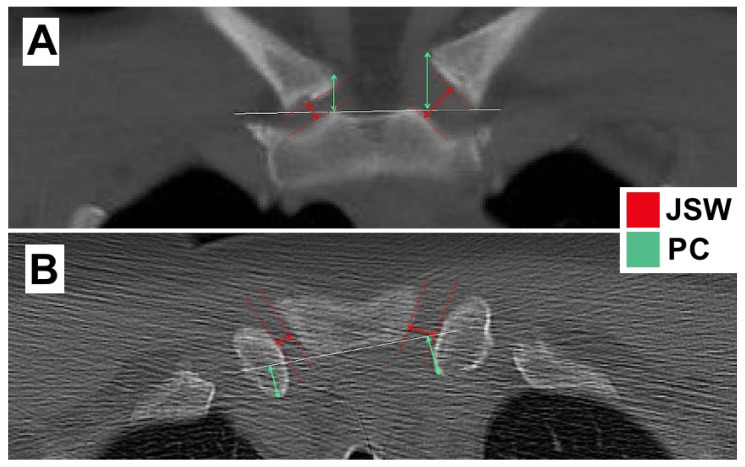
(**A**,**B**). CT scan in coronal (**A**) and axial (**B**) plane for measurement of the joint space width JSW (red) and position of medial clavicle PC (green). The JSW was determined by the least distance between the clavicle and sternal corticalis. PC, in contrast, was determined by the distance between the apical sternal corticalis and the apical end of the medial clavicle in the coronal plane, and by the posterior corticalis of the sternum and the posterior end of the medial clavicle in the axial plane.

**Figure 2 ijerph-19-15011-f002:**
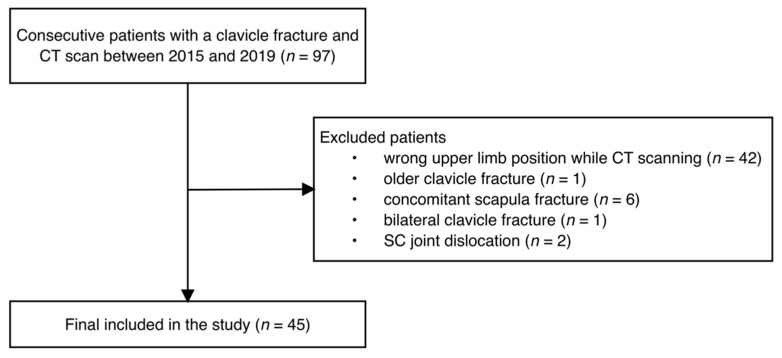
Flow chart showing the definition of the study population.

**Table 1 ijerph-19-15011-t001:** Clavicle fracture characteristics.

Robinson Classification	*n (%)*
Medial	4 (8.9)
*1A1*	-
*1A2*	2 (4.4)
*1B1*	-
*1B2*	2 (4.4)
Midshaft	31 (68.9)
*2A1*	8 (17.8)
*2A2*	3 (6.7)
*2B1*	11 (24.4)
*2B2*	9 (20.0)
Lateral	10 (22.2)
*3A1*	3 (6.7)
*3A2*	-
*3B1*	5 (11.1)
*3B2*	2 (4.4)
Total	45

*1A1* medial, undisplaced, extra-articular. *1A2* medial, undisplaced, intra-articular. *1B1* medial, displaced, extra-articular. *1B2* medial, displaced, intra-articular. *2A1* midshaft, cortical alignment, undisplaced. *2A2* midshaft, cortical alignment, angulated. *2B1* midshaft, displaced, simple or wedge comminuted. *2B2* midshaft, displaced, isolated or comminuted segmental. *3A1* lateral, cortical alignment, extra-articular. *3A2* lateral, cortical alignment, intra-articular. *3B1* lateral, displaced, extra-articular. *3B2* lateral, displaced, intra-articular.

**Table 2 ijerph-19-15011-t002:** Joint space width and position of the medial clavicle in affected and non-affected side.

	cJSW	aJSW	cPC	aPC
Affected side in mm	8.70 (4.30–15.50)	8.98 (4.50–4.10)	14.31 (5.50–31.30)	9.40 (2.20–14.40)
Non-affected side in mm	7.63 (3.40–13.40)	8.32 (4.10–13.20)	13.49 (4.90–22.40)	9.02 (1.00–9.02)
Mean difference in mm	1.07 (1.67)	0.67 (2.02)	0.83 (2.07)	0.38 (2.20)
Mean difference in %	14.02 m	7.93 m	6.08 m	4.02 m
*p*-value	<0.001	0.044	0.011	0.274

Dimensions of mean SC joint space width (JSW) and position of the medial clavicle (PC) in coronal (c) and axial (a) imaging plane. Data are given as mean ± standard deviation and range.

**Table 3 ijerph-19-15011-t003:** Difference of affected vs. non-affected side for joint space width and position of the medial clavicle in relation to clavicle fracture location.

Difference of Affected vs. Non-Affected Side for	Clavicle Fracture Location			*p*-Value
	Medial (*n* = 4)	Midshaft (*n* = 31)	Lateral (*n* = 10)	
cJSW	3.30 ± 1.37	1.31 ± 1.36	1.63 ± 0.96	0.048
aJSW	3.17 ± 2.74	1.32 ± 1.41	1.61 ± 1.30	0.547
cPC	1.34 ± 0.89	1.72 ± 1.19	2.10 ± 2.09	0.884
aPC	0.40 ± 0.35	1.9 ± 1.77	1.31 ± 1.44	0.547

Differences of affected vs. non-affected side of mean sternoclavicular joint space width (JSW) and position of the medial clavicle (PC) in coronal (c) and axial (a) imaging plane depending on clavicle fracture location. Data are given in mm as mean ± standard deviation and range. The Kruskal–Wallis test was used to compare the mean values across the three clavicle fracture locations (medial, midshaft, lateral). Mean values were calculated from the difference between the affected and non-affected sides.

## Data Availability

Not applicable.

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
