# Peer review of "Morphologic Alterations of the Sternoclavicular Joint following Ipsilateral Clavicle Fractures"

_ijerph, 2022, doi:10.3390/ijerph192215011_

Round 1

Reviewer 1 Report

The manuscript “Morphologic alterations of the sternoclavicular joint following ipsilateral clavicle fractures” by  Malik Jessen et al. aimed to determine the effect of ipsilateral clavicle fracture on the sternoclavicular joint (SCJ) in a systematic computer tomography (CT) morphologic evaluation

Below are my comments and remarks regarding the manuscript:

1. No introduction in the abstract

2. Introduction - The first sentence that defines a sternoclavicular joint (SCJ) is redundant

3. Methods lacks a description of Robinson classification and justifications for its application

4. Methods did not specify who and with what experience performed the radiographic measurements

5. Fig. 1 AB, the abbreviations JSW and PC should be added in the figure to make it easier to illustrate

6. The mean value of the 4 cases in the medial group is completely statistically unreliable and no conclusions should be drawn on the basis of such a small group. Only midshaft vs lateral should be compared or the number of medial cases should be increased and not included in the discussion.

Author Response

Dear Sir or Madam,

Thank you both for the kind and detailed review of our manuscript. Enclosed are our responses to the comments mentioned by the reviewers:

REVIEWER #1

Reviewer:

No introduction in the abstract.

Authors' response:

We have added a short sentence to the introduction right before the rationale.  We are happy to add more if you deem appropriate (line 13-14).

Reviewer:

Introduction - The first sentence that defines a sternoclavicular joint (SCJ) is redundant.

Authors' response:

We have deleted "(SC)" in the first sentence (line 32).

Reviewer:

Methods lacks a description of Robinson classification and justifications for its application.

Authors' response:

We have added further information on Robinson Classification in the following text passages: introduction section (line 67 -71), method section (line 100-101), figure text (Iine 164-170).

Reviewer:

Methods did not specify who and with what experience performed the radiographic measurements.

Authors' response:

To provide clarification, we have added the necessary information to the methods section (line 99-100). As already mentioned in the discussion section (line 308), only one experienced physician performed the evaluation.

Reviewer:

Fig. 1 AB, the abbreviations JSW and PC should be added in the figure to make it easier to illustrate.

Authors' response:

We have added the abbreviations in the figure for a better and faster understanding (figure 1 in line 131).

Reviewer:

The mean value of the 4 cases in the medial group is completely statistically unreliable and no conclusions should be drawn based on such a small group. Only midshaft vs lateral should be compared or the number of medial cases should be increased and not included in the discussion.

Authors' response:

Indeed, a cohort size of only 4 cases is very small and conclusions should be drawn with caution. We have therefore clearly pointed out this fact in the limitations of the study (line 315-317). If this is not sufficient for you, we can also change the relevant text passages where significant results regarding the medial clavicle fracture are referred to. An isolated increase in the number of cases of medial clavicle fractures will be difficult.

Reviewer 2 Report

Thank you for the opportunity to review your manuscript, Morphologic alterations of the sternoclavicular joint following ipsilateral clavicle fractures.

Some of the references used are too old (number 6), and I consider that they do not bring anything new to the table.

Was it the same evaluator who made the measurements? From your comment on limitations, we understand that yes. In limitations, you indicate the reliability of another study that is not cited. I think that the intra-observer reliability of the study itself should be performed using a small sample of images selected and measured several times by the same evaluator. I think it would add more value to the article.

Line 105. Are the subjects 97 or 98 as stated in the flow chart?

Table 1 discusses Robinson's classification. I believe this classification should be introduced briefly beforehand to understand the table better.

The tables are generically named at the top, and indications for better interpretation are given at the bottom.

Line 121. The term " appeared" generates confusion. 

Line 124. Significant difference - do you mean statistically significant difference? If so, you must give the value even if it is in the table. It makes it easier to read. 

Table 3. The heading of the table is inadequate. The table should be named briefly at the top, and the explanation necessary for understanding should be at the bottom.

Line 144 - 163 provides information on the pathology but does not discuss the findings.

Line 229-230. The content of this line should be left for discussion. It cannot be affirmed based on the present study.

Author Response

Dear Sir or Madam,

Thank you both for the kind and detailed review of our manuscript. Enclosed are our responses to the comments mentioned by the reviewers:

Reviewer:

Some of the references used are too old (number 6), and I consider that they do not bring anything new to the table.

Authors' response:

We have removed some old references respectively replaced them with newer citations.

Reviewer:

Was it the same evaluator who made the measurements? From your comment on limitations, we understand that yes. In limitations, you indicate the reliability of another study that is not cited. I think that the intra-observer reliability of the study itself should be performed using a small sample of images selected and measured several times by the same evaluator. I think it would add more value to the article.

Authors' response:

Exactly, it was one shoulder fellowship trained physician, we added this information in the methods section (line 99-100). This is exactly how we determined the intraobserver reliability in our preliminary work (line 308-312).

Reviewer:

Line 105. Are the subjects 97 or 98 as stated in the flow chart?

Authors' response:

Absolutely correct, 97 patients in total (as described in the text and as also further calculated in the flow chart). Of course, we have corrected it in the first box in the flow chart from 98 to 97 patients.

Reviewer:

Table 1 discusses Robinson's classification. I believe this classification should be introduced briefly beforehand to understand the table better.

Authors' response:

We have added further information on Robinson Classification in the following text passages: introduction section (line 67 -71), method section (line 100-101), figure text (Iine 164-170).

Reviewer:

The tables are generically named at the top, and indications for better interpretation are given at the bottom.

Authors' response:

We have adjusted all tables accordingly.

Reviewer:

Line 121. The term " appeared" generates confusion. 

Authors' response:

We have replaced the word "appeared" with " was" (line 179).

Reviewer:

Line 124. Significant difference - do you mean statistically significant difference? If so, you must give the value even if it is in the table. It makes it easier to read. 

Authors' response:

The p-value is already given: “The position of the clavicle showed a significant difference in the coronal plane indicating a superior shift of the clavicle after clavicle fracture (cPC; mean difference 0.83mm [6.08%]; p = 0.011). “

However, we have moved the p-value forward in the sentence for easier reading (line 183).

Reviewer:

Table 3. The heading of the table is inadequate. The table should be named briefly at the top, and the explanation necessary for understanding should be at the bottom.

Authors' response:

We have adjusted all tables accordingly.

Reviewer:

Line 144 - 163 provides information on the pathology but does not discuss the findings.

Authors' response:

You are right, it makes more sense to move the information to the introduction section (line 43-65).

Reviewer:

Line 229-230. The content of this line should be left for discussion. It cannot be affirmed based on the present study.

Authors' response:

We agree and have therefore removed the sentence from the conclusion (line 325-328).

Round 2

Reviewer 1 Report

Last point: Please change the relevant text passages where significant results regarding the medial clavicle fracture are referred to. The more so as the statistical significance is on the border of 0.05

Author Response

Reviewer:

Last point: Please change the relevant text passages where significant results regarding the medial clavicle fracture are referred to. The more so as the statistical significance is on the border of 0.05.

Authors' response:

We have made the following changes regarding the medial clavicle fracture:

  • Line 198-201: We rephrased the results regarding the significant finding of the cJSW.
  • Line 210: The sentence has been deleted.
  • Line 269-272: We underlined that we do not attribute any effect to the significant finding.
  • Line 341-343: It was already added in the last revision.

Of course, if needed, additional changes beyond the above-mentioned aspects can be made.

Reviewer 2 Report

The authors have answered all my questions.

I believe the article is suitable for publication. 

However, there is one minor unresolved issue. There is a mismatch between the initial sample of 97 subjects and the 98 listed in the FlowChart. The authors have not corrected this error.

Author Response

Reviewer:

The authors have answered all my questions.

I believe the article is suitable for publication.

However, there is one minor unresolved issue. There is a mismatch between the initial sample of 97 subjects and the 98 listed in the FlowChart. The authors have not corrected this error.

Authors' response:

We appreciate your feedback. Indeed, we have adjusted the graphic accordingly, but we missed to properly transfer it into the revised manuscript. Please notice that now the updated graphic can be found in the manuscript (line 163).